# Post-Migration Stressors and Mental Health for African Migrants in South Australia: A Qualitative Study

**DOI:** 10.3390/ijerph19137914

**Published:** 2022-06-28

**Authors:** Lillian Mwanri, Nelsensius Klau Fauk, Anna Ziersch, Hailay Abrha Gesesew, Gregorius Abanit Asa, Paul Russell Ward

**Affiliations:** 1Research Centre for Public Health Policy, Torrens University Australia, Adelaide, SA 5000, Australia; nelsensius.fauk@torrens.edu.au (N.K.F.); hailay.gesesew@torrens.edu.au (H.A.G.); gregorius.asa@student.torrens.edu.au (G.A.A.); paul.ward@torrens.edu.au (P.R.W.); 2Institute of Resource Governance and Social Change, Kupang 85227, Indonesia; 3College of Medicine and Public Health, Flinders University, Adelaide, SA 5001, Australia; anna.ziersch@flinders.edu.au; 4College of Health Sciences, Mekelle University, Mekelle P.O. Box 231, Tigray, Ethiopia

**Keywords:** mental health challenges, African migrants, refugees, risk factors, post-migration stressors, Australia

## Abstract

We conducted a qualitative study involving African migrants (*n* = 20) and service providers (*n* = 10) in South Australia to explore mental health stressors, access to mental health services and how to improve mental health services for African migrant populations. This paper presents the views and experiences of African migrants about the post-migration stressors they faced in resettlement that pose mental health challenges. The participants were recruited using the snowball sampling technique. To align with the COVID-19 pandemic protocol, the data collection was conducted using one-on-one online interviews through Zoom or WhatsApp video calls. Data analysis was guided by the framework analysis. The post-migration stressors, including separation from family members and significant others, especially spouses, imposed significant difficulties on care provision and in managing children’s attitudes and behavior-related troubles at school. African cultural practices involving the community, especially elders in care provision and disciplining children, were not consistent with Australian norms, compounding the mental health stressors for all involved. The African cultural norms, that do not allow young unmarried people to live together, also contributed to child–parent conflicts, enhancing parental mental stressors. Additionally, poor economic conditions and employment-related difficulties were post-migration stressors that the participants faced. The findings indicate the need for policy and intervention programs that address the above challenges. The provision of interventions, including social support such as subsidized or free childcare services, could help leverage their time and scheduled paid employment, creating time for effective parenting and improving their mental health and wellbeing. Future studies exploring what needs to be achieved by government and non-governmental institutions to support enhanced access to social and employment opportunities for the African migrant population are also recommended.

## 1. Introduction

The Department of Economic and Social Affairs (DESA) of the United Nations Secretariat defines migration as a process where a person or populations move from one cultural setting to another to settle for a longer period or permanently [1]. The 2020 World Migration Report provides evidence suggesting higher rates of mental distress among refugee, asylum seekers and migrant populations [2]. Globally, the prevalence of mental health issues, including stress, anxiety, depression and post-traumatic stress disorders, are commonly reported to be higher among these populations compared to host populations [2,3,4,5,6]. A recent systematic literature review and meta-analysis, involving 44,365 migrants in 17 countries, reported an overall prevalence of depression and anxiety, accounting for 38.99% and 27.31%, respectively [7]. Studies in Australia have correspondingly reported higher mental illness among migrant populations compared to non-migrant Australians [8,9,10,11,12].

The experience of a wide range of stressors associated with the migration journey, including pre-migration, and during migration, settlement and integration in host countries have been reported as risk factors for mental health issues [2,5,13]. Pre-migration experiences, including exposure to a range of traumatic events, such as violence, persecution, war or civic unrest, torture and imprisonment in their home countries, contribute to the mental health challenges [14,15,16]. Such stressful events not only affect their mental health condition at the individual level, but also influence their relationships with families and communities as they are often forced to leave their home countries, even without basic living essentials, such as food, water and shelter [2,5,16,17]. Their experiences of difficulties during the migration process and the length of stay in detention camps with limited or lack of access to services and social support have also been reported as stressors for mental health issues [18,19,20].

Post-migration stressors, such as family separation, worry about family back home and inability to visit family in their home countries, have been reported to be significantly associated with anxiety, depression and post-traumatic stress disorder among the migrant populations [18,21,22]. Similarly, difficulties with social integration and weak social networks in the host countries can contribute to mental health issues, such as stress, anxiety and depression among migrants [23,24]. Their experiences of issues in their new host countries, including difficulties in language learning, unemployment, financial difficulties and difficulties in understanding the new systems and complex settlement processes to access essential services, are additional factors that contribute to poorer post-migration mental health [5,12,25,26,27,28,29]. Language barriers, for example, have been reported as a significant influence on employment and subsequent mental health [5,30]. Unemployment and financial issues, which also lead to difficulties in affording housing and/or accommodation, have also been reported as post-migratory factors that increase the risk of mental health issues among the migrant populations [18,24,31,32]. Additionally, discrimination has also been reported as a contributor to resettlement stressors, impacting the mental health of these populations in the host countries [5,31,33,34].

Corresponding to findings elsewhere, the studies involving the migrant populations in Australia have reported poor social integration, concerns about families left behind and loneliness as common post-migration stressors associated with mental health issues in resettlement [8,9,17,35,36]. Moreover, unemployment and financial stressors have been associated with poor mental health outcomes among the migrant populations in Australia [14,16,37]. Furthermore, it has been suggested that the unemployed migrants, compared to the migrants who are employed in Australia, have higher levels of psychological distress [38]. Unemployment and financial difficulties have also been reported to be connected to poverty, housing instability and poor access to healthcare services, further complicating post-migration mental health among the migrants in Australia [9,10,12,39]. Some studies have also reported racial discrimination as an important post-migration risk factor for high psychological distress among the migrants in Australia [40,41,42,43].

Despite a range of risk factors for mental health issues among the migrant populations and refugees globally, and in the context of Australia as reported in the aforementioned studies, there is a lack of evidence on how post-settlement factors, such as spousal relationships, children-related issues, cultural practices of raising children, and issues related to mortgage, loans and bills, influence the mental health outcomes of the migrant populations in the host communities. This paper aimed to fill in this gap in the knowledge and focused on exploring post-settlement stressors for mental health issues among African migrants in South Australia. In particular, the attention is focused on the family and community relationships, parenting and financial matters. Understanding these factors can be useful for the development of policy and intervention programs to support migrant populations’ resettlement and integration in Australia, and in other settings globally.

## 2. Methods

We provide a brief overview of the methods in this section, as the details have been previously reported elsewhere [44,45].

### 2.1. Study Design and Participant Recruitment

We used a qualitative study design to explore the views and experiences of 20 African migrants and 10 service providers in South Australia regarding the issues related to post-migration stressors for mental health challenges, the barriers to accessing mental health services and the strategies to improve the access of African migrants to the services [44,45]. The use of qualitative design in this study assisted us to explore in-depth the participants’ views, experiences, understanding and interpretations of various factors that influenced their mental health condition after their arrival in Australia. Out of this broader study, the current paper presents the perspectives and experiences of the 20 African migrants regarding the post-migration stressors affecting mental health.

The recruitment used a snowball sampling technique. The researchers started the recruitment by distributing the study information packs to potential participants through community groups or associations, and organizations providing services for the migrant population in South Australia. The information packs contained contact details of the field researcher. The potential participants who received the information packs personally contacted the field researcher to confirm their participation and were recruited for an interview. Following the interviews of the initial participants, the snowball sampling technique was also employed, by asking them to disseminate the information packs to their families, or eligible friends and colleagues who might be willing to take part in this study. This procedure was used throughout the recruitment process, and finally, 20 African migrants were recruited and interviewed. None of the participants withdrew from the study prior to or during the interviews. None of the participants were known to the researchers prior to the recruitment and interviews. The inclusion criteria used to recruit the participants were: (i) the participant was an African migrant; (ii) the participants were aged 18 years old and above (for the community participants); or (iii) the participant was a member of staff at an institution or organization providing services for migrants and/or refugees (for the service providers).

As the study was based on voluntary participation, it does not represent the targeted population, rather it provides rich and detailed data of the experiences of the participants. The recruitment was also through snowball methods which makes it difficult to estimate how many people received the information but did not volunteer to participate. Of the 20 African migrants included in the study (half female and male), the age interval ranged from 18 to 60 years old, with the majority being aged 41 to 60 years (*n* = 12 people). They were originally from eight different African countries, namely: The Democratic Republic of Congo; South Sudan; Liberia; Sierra Leone; Burundi; Ethiopia; Kenya and Somalia. They were self-identified as African migrants. The duration of their living in Australia ranged from 3 to 20 years. The majority of the participants (*n =* 11) engaged in different types of paid jobs, while the others (*n =* 7) were currently unemployed or actively searching for a job, and the remainder (*n =* 2) were full-time university students. The participants were of a mixed background, with the majority being immigrants, a couple being refugees and none being an asylum seeker. They are referred to as migrants, as our focus was mainly on post-resettlement stressors for mental health challenges facing them, regardless of their status or how they arrived in Australia.

### 2.2. Data Collection and Analysis

One-on-one in-depth interviews were carried out from May to October 2020 to collect the data from the participants. The interviews were conducted using online platforms (Zoom or WhatsApp video calls), which were mutually agreed upon by the participants and field researcher, due to the COVID-19 related protocols and restrictions. The duration of the interviews ranged from 40 to 60 min. The interviews were recorded using a digital recorder and conducted in English. Prior to the interviews, each participant was asked whether they required an interpreter to assist them but none of them requested an interpreter. The participants could speak English and engaged well with the conversation during the interviews. Fieldnotes were also sometimes undertaken by the interviewer, if it was felt necessary, and then integrated into each transcript during the transcription process. The interviews ceased once no new added information or response was provided by the last few participants, which was an indication of data saturation. All of the interviews were conducted by a female researcher from a non-African background who has significant experience in conducting research activity in these populations. There was no repeated interviews with any of the participants. At the end of each interview, the interviewer offered an opportunity for each participant to read and comment on the interview transcript after transcription, but none of the participants requested to do so.

The data were transcribed verbatim, and the analysis was guided by the five steps of qualitative data analysis framework, introduced in Ritchie and Spencer [46,47], which include: (i) familiarization with the data. This was carried out through repeatedly reading the transcript of each interview, breaking down the information into small chunks of information or data extracts, and highlighting and giving comments and labels to those data extracts; (ii) identification of a thematic framework, through which the recurrent and emerging key ideas and concepts regarding the topic under research from each of the interview transcripts were identified and listed. These ideas, issues and concepts were then used to develop the thematic framework. The thematic framework was identified and developed deductively, based on prior knowledge, and inductively based on the emerging themes; (iii) indexing or coding of the data, which was started by creating open codes for the data extracts that had been created in the previous steps, resulted in a collection of a long list of codes. Following this, close coding was carried out, to identify and collate the similar codes and to reduce the number codes to a manageable number. In addition, the different codes that formed a pattern were grouped under the same theme or sub-theme; (iv) creating a chart for the data, through reorganizing the themes and their data extracts that had been created and coded in the previous steps in a summary chart, which enabled a data comparison within each interview and across the interviews; and (v) mapping and interpretation of the data entirely [44,45]. These steps were followed to support the management of these qualitative data in a structured way and to enhance the rigor and transparency of the analytical process [46,48,49]. We have reported other findings from the study elsewhere [44,45].

Each of the participants signed an informed consent form to confirm their willingness to participate, and they returned the consent form to the interviewer prior to the commencement of the interviews. This study was approved by the Social Science and Behavior Research Ethics Committee, Flinders University (No. 8570). The details about the ethics approval and the consent from the participants have been previously reported elsewhere [44,45].

## 3. Results

The analysis identified three key themes relating to post-migration stressors affecting mental health, namely: family-related factors; culture-related factors; and economic factors, and one further key theme about community strengths. The thematic details alongside the subthemes are described below.

### 3.1. Family-Related Factors

#### 3.1.1. Family Disconnection

Disconnection or separation from family members was reported as having a significant negative influence on the mental health (e.g., stress, anxiety) of both the female and male participants. The participants described how they had been challenged and stressed as a result of leaving family members behind in Africa, not knowing or receiving information about them and trying to bring them to Australia. Upon arrival in Australia, they also became anxious about not knowing what the future held. The following narratives of two women who had been living in Australia for 20 and 12 years, respectively, reflect some of the mental health issues that the participants suffered, due to the conditions they faced post-migration and settlement period in Australia:


*“I was stressed out, not only anxiety of not knowing what was going on with the rest of my family (back in Africa), but separation from them also brought on stress on me. I felt it quite stressful and at that time I actually …. I was so anxious about my family back at home, oh, but at that time there was nothing which I could do here, it was so desperate”.*
(P5, male, 50 years old)


*“Yeah, but there was a lot of trouble when I came. I came without my kids. I had to try very hard to bring them out together. And it was really, really challenging and stressed me out”.*
(P9, female, 50 years old)

The mental health challenges (e.g., depression, stress and feelings of loneliness) seemed to be exacerbated by the fact that many did not know anybody in Australia during their early period of arrival and had few social connections. Such situations were described as leading to their experience of loneliness. They also missed their friends and family, leading to further stress and social isolation. These situations are illustrated in the following narrative of a young woman, who at the time of the interview had arrived in Australia with her mother three years earlier, leaving her father and sibling in her home country:


*“During the early period of living in Australia, I was really depressed because I was missing my friends. They’re back in Africa. And I didn’t have a phone so I can’t phone them. And they didn’t have a phone too. So, it was really hard for me. And I didn’t know anyone here. Most of the time I am just sitting in my room, lying on my bed, doing nothing …. I was starting a new life and it was really difficult for me to know anyone or make friends because I’m not the kind of person who can make friends easily, …. It’s usually affecting my mood, I was stressed, and also the way I behave around people. I’ll be feeling lonely when I’m around people. Yeah. Yeah. Especially to the people—I don’t know. I didn’t know anyone, so it was weird. And I didn’t feel comfortable sometimes, around people”.*
(P17, female, 20 years old)

Difficulties in socializing with other people or friends at schools or within the communities where they lived during the resettlement period were also a stressor for the mental health issues or stress facing some of the female participants. Stories of some of the women showed that this led to self-isolation and feeling lonely, uncomfortable around other people and stressed:


*“I usually don’t do anything apart from just sitting in my room laying on my bed doing nothing. So, I used to just do nothing, so I used to hand my school work in late. It used to make me sit far from people, so I didn’t communicate with people”.*
(P20, female, 18 years old)


*“Especially when I was in school. I didn’t know anyone, so it was weird. And I didn’t feel comfortable sometimes, around people. Yeah. …. It’s usually affecting the mood, my mood, I’ll be stressed. And also the way I behave around people. I’ll be feeling lonely when I’m around people”.*
(P17, female, 20 years old)

#### 3.1.2. Spousal Relationships and Children Related Issues

Spousal relationship issues were indicated to negatively impact on some of the female participants’ mental health outcomes. For example, three of the female participants reported being abandoned by their husband or fiancé. Their expectations for their own families and relationships seemed to be ruined by such separation or break up, which further caused them various negative psychological challenges, such as stress and feeling of loneliness. It also seemed that the COVID-19 situation and the enforcement of the preventive protocols may have corroborated these challenging mental health issues that they faced. The following narrative of a woman, who has been living in South Australia for more than ten years, illustrates these assertions:


*“I have a lot of problems with my health. Especially, right now I’m sitting—I’m talking with you as a human, but I have a deep thing in my heart that I really need to let out. Since the COVID-19, it has been disturbing me a lot. And can you imagine, you came with a family and because of money or some kind of …. your husband just left you all alone by yourself with the kids, yeah. And it has been bothering me a lot. …. And right now personally as I’m talking with you, I have a counsellor who’s supposed to be meeting me tomorrow, so we can talk about it. Because I’m stressed out. Can you imagine you expect your family to be a certain way, and things are not working out for you? Like, for me, I got married back home, and all my family knows that I’m with my husband and I came here and he’s no more with me. It stresses me out a lot”.*
(P16, 54 years old)

Spousal relationship issues seemed to have consequences on the participants’ responsibility to raise their children. Similar stories were echoed by other women who were sole parents, who found it difficult to raise the children on their own. For example, a female participant described how her children had experienced violence from their father which led to their separation, but because of the lack of resources or support, she at times had to seek his support to look after their children as she needed to work. This situation was described as painful to her, as she did not want to put her children into such a risky environment:


*“We (the woman and her husband) used to live together, but unfortunately we separated due to violence. So right now they (my children) live with me but sometimes I take them to their dad when I’m going to work because nobody’s there to look after them and I don’t want to make other people feel like I am bothering them with my kids. The fact that we separated, I normally just say—it pains me a lot to take them (my children) back to the environment, because he (my former husband) drinks a lot and it’s not safe for them, it pains me a lot. But at the same time, I don’t have any other options. I moved out from everything, but then the kids make me go back”.*
(P11, 37 years old)

Concerns about children’s attitudes and behaviors at schools were acknowledged to be a source of increased stress and frustration by the participants. Because of poor behavior at school, teachers were unhappy and the impact of these behaviors led to more complex social issues. The stories of both male and female participants presented below illustrate the concerns that the participants had regarding their children’s educational performance:


*“I will start with the children because in so many schools our children are facing so many problems, you know? Every now and then teachers expel them from class due to misbehaving. They say and look at the teacher some kind of way, or respond to the teacher some kind of way, or ask some questions some kind of way. …. He (his son) was using his phone in the class, you know, all sorts of things, …That’s another stress”.*
(P18, male, 45 years old)


*“It is sometimes frustrating because he (her son) went to school but sometimes didn’t attend the class. I knew about it once I got a call from the teacher”.*
(P6, female, 37 years old)

### 3.2. Cultural Parenting Practices

The cultural differences in parenting between Africa and Australia were also mentioned to be challenging for the participants. The collectivist cultural practices regarding how people in Africa raised their children seemed to have an impact on the mental health of the participants, where these practices were inconsistent with the expectations in Australia. For example, the practice in Africa where children were disciplined by elders within the community members was not the norm in Australia. The community support in parenting that was available in Africa was perceived as safe, as parents could leave young children without an adult when they went out for work or for other reasons. Because such cultural practices were not acceptable in Australia, the lack of support made it difficult for the participants to sufficiently take care of their children, further exacerbating their mental health issues:


*“In Africa, you will just leave your kids at home and they will be looked after because kids are being raised by villagers, and villagers are there to take care of your kids. We just take care of the people living near us and around us. When I came to Australia, I found it different, because at the age of six we never take our kids to school. They go by themselves. We leave them at home and we just go to the market, we go to fetch water, and we leave them home”.*
(P19, female, 29 years old)


*“You can now see a lot of mental illness is happening. It is also because the role of the elders back home is not like here. Back home the role of the elders is that if an elder goes out and sees something going wrong with the young person then the elder will discipline the child because for Africans the child belongs to the community. So, like my children there, if they go out, any African out there will look after them and discipline them if they see them doing something wrong because children belong to the community. And this is where it goes wrong here. Here it is a very individualised society where everything is about themselves. And this is where a lot of things are going astray. And you can see a lot of kids that are doing all the bad stuff because they know no one will stop them”.*
(P1, male, 52 years old)

The differences in the cultural practices regarding children’s relationships seemed to also become a concern for some of the parents and brought stress. For example, children’s relationships with the opposite sex at a certain age, were unacceptable to some of the parents, as it is not allowed in their African culture and was described as a cause of both stress and child–parent conflict. The following story of a 50-year-old man who had been living in Australia for 12 years reflects such a situation:


*“I am having a problem with my boy. He was misbehaving, yeah. The boy is trying to find a place for himself. Well, I’m still looking for a place for him now. For yeah, he was giving me a lot of stress during that time, because when he grows up, he was trying to bring—I mean, girlfriend in the house. In our culture, we don’t do that, you know? I mean, you have to respect your parents. But here, you don’t respect your parents, yeah. Everything, you do as you want. As soon as you say, “No, this is not right.” He said, “Okay, I’m going to find my own place.” Yeah, so I decided now I’m trying to get a place for him. A room and that would be nice for him so that he can settle down. Yeah, but for now, he’s living with his girlfriend, yeah. I said (to the girl), go away and go live with your girlfriend, that’s fine. Yeah, you see”.*
(P9)

### 3.3. Economic Factors

Meaningful employment is an important mental-health protection for wellbeing, successful settlement and integration. Poverty and financial difficulties were experienced and indicated as a vicious risk factor for mental health issues, such as stress, that both the female and male participants faced. These were reflected through the difficulties in fulfilling the necessities for their family, including the inability to pay school fees for their children. The following story of a 37-year-old woman reflects the economic or financial hardships that the participants had experienced, which negatively impacted their mental health:


*“It is financially hard, I have to cook for them (her seven children), I have to clean for them. I have to do everything for them. Nothing with the hospital or nothing with the community. I am the one who provides for my seven children, not my husband. What they can eat, what they can buy, their school fees or needs, and all this. I am struggling and this is very stressful”.*
(P11)

The inability to pay loans and bills was another reflection of the economic or financial difficulties, which seemed to be the consequences of the difficulties for these participants to secure jobs in Australia. All of these difficulties seemed to be a source of significant disadvantage and negatively impacted on their mental health, such as stress and depression. This was also exacerbated by the impact of the pandemic that the world has experienced during the past two years. It is undeniable that COVID-19 has caused a massive lay off in many sectors and the loss of job opportunities for many people, and this situation certainly contributed to the difficulties for African migrants to find a job. These assertions are illustrated by the following accounts of participants’ reflections below:


*“I have issues with mental health and it is also because of the stress that I am getting from loans, bills—now if you look at—because of the corona, I am not getting money, that all create another other stress, a mental health, for me. …. I think things will be all right, but things are moving too fast that I cannot cope all the time, I cannot pay the bills on time. You can imagine—there are so many things”.*
(P18, male, 45 years old)


*“And because I cannot get a job or get anything to do, I have to depend on my daughter to do half of the things for me. Sometimes, before I finish paying my bills, I have nothing left in my hand, and I have to wait for another two weeks before I can get something. You see, all these things, all I think about and they depress me a lot, I won’t lie. I am very, very depressed”.*
(P16, female, 54 years old)

Meaningful employment can shape the migrants’ lives and establish basic needs, such as housing. For some, owning a home was a dream for a successful resettlement, but having a mortgage also seemed to be a cause of psychological burden, such as stress due to the inability to repay. For example, the following story of a 54-year-old man who had been living in Australia for 16 years illustrates how mortgage stress had made him scared and sleepless, due to over thinking about the ways to repay the loan:


*“Then you see, again in Africa, we have our own house. We don’t have to pay for it all the time, all of that. When we came here, we try and get like a mortgage. If you have a mortgage, you have to pay for it. This makes me live under stress because I don’t have a stable job. It’s just scary, how am I going to pay for this house in 30 years? I am not sleeping more because of that”.*
(P3)

Some of the participants described that they found it hard to secure a job, due to various reasons. While for some, the lack of work experience was the cause of job insecurity, for others, poor English language proficiency put them at increased odds of being unemployed, which was a source depression and frustration:


*“When we were in Africa and we heard about white man land, we just feel that when we come, everything is going to be okay. Everything is going to be fine. Everything is just going to be all smooth, on a silver platter. But I came, and it’s not like that. You have to earn a living. I was really, really, really depressed and like, what happened? Because I apply (for a job) from place to place, and they wanted someone with working experience in Australia and I don’t have working experience in Australia, so it was very difficult, and I was really frustrated about it”.*
(P7, female, 41 years old)

For some of the participants, the background of being refugees had left their bodies battered. For the fifty-year-old male below, despite his desire to work, earn and make a living in Australia, physical weakness acted as a barrier to gaining employment and was a source of frustration and further stress:


*“Unfortunately, I could not get a job. …. I get my back problem, neck problem, then my vision. …. I know I do not have that power to work, but I want to work. I know I don’t have that (physical) strength to go to work. Very frustrating, it’s very, very hard”.*
(P5, male, 50 years old)

### 3.4. Community Strengths

Despite the mental health challenges that the participants faced, the strong ties with their communities reflected the potential strength that could be employed as a strategy to help them cope with their mental health challenges. Most of the participants acknowledged their association with the community from their country of origin, which showed strong social and cultural bonds among them, as reflected in the following quotes:


*“I am associated with the Somali communities. I have friends from other communities but I’m always with the Somali community”.*
(P8, female, 19 years old)


*“The community I associate with is my community, the Congolese community. I have been here for many years (15 years) so I know many people in our (Congolese) community. I mainly associate with them ….”.*
(P5, male, 50 years old)

The regular social, cultural and religious events and festivals within the communities and churches seemed to be the instruments that facilitated the participants’ or community members’ connection with each other. The participants’ and their community members’ regular engagement in such events also showed their communal characteristics, which could be amplified to facilitate help and support for one another within their communities, including support for the ones facing mental health issues:


*“I usually associate with the South Sudanese community. We have cultural and social events, and meetings regularly within our community. We regularly meet at the church during Sunday mass or other religious events. Lately, we haven’t had any events or any special meetings because of the COVID. The church was closed, and I’ve never been to the church again, after the closure”.*
(P17, female, 20 years old)


*“Q: …. Which communities do you mostly associate with?*



*A: Because I live in the Salisbury area, so most of the time I go to Elizabeth, and Munno Para. Yeah, yeah. That’s the community that’s closer to me so most of the time I go Elizabeth, Munno Para, Parafield Gardens.*



*Q: Okay. And do you ever go to events and things like that with the African Women’s Federation, or with the Liberian community, or anything like that?*



*A: Yes. We always have festivals and then we meet. We meet, and we have fun. We always meeting as a group”.*
(P14, male, 42 years old)

## 4. Discussion

In Australia today, migration has meant the population has become highly diversified with increases in the arrivals of migrants and refugees, including from Africa. Migrant populations can be a highly vulnerable group in terms of postmigration stressors, that can have detrimental effects on their mental health and other outcomes [2,16,18]. In this paper, we have identified some of the post-migration mental health stressors faced by African migrants to Australia.

The stressors identified, relating to family relationships, cultural parenting practices and the economic or financial difficulties faced by these migrants in Australia, are complex and ongoing. The stressors accounted for by the participants are real and impact on their successful resettlement and integration in Australia. As described by some of the participants, family-related issues, including broken spousal relationships and abandonment by a fiancé, were key post-migration stressors that affected the mental health of some of the female participants. While breakups are not unique to these populations, breakups are emotionally taxing because of the perceived rejection, as well as the perception of it being a shame to the family [50,51,52]. The rejected partner, often the female, experiences grief, loss and sadness. Additionally, and for these populations in particular, the loss of spousal, or a partner’s, support means a lot, due to the poor resources, and may amplify the existing difficulties in coping with daily life and family needs, increasing the psychological burden. Spousal separation or divorce has been reported to have an impact on children, including being unhappy and stressed which can translate into low self-esteem, behavioral problems and a sense of loss [53,54]. Such situations can exacerbate the difficult pre-migration situations, caused by the separation from families and friends left behind in their home countries, which were reported by all of the participants. These findings are in line with previous studies, with African migrants and migrants in general in other settings, globally and in the Australian context [2,9,18,37,55], reporting the disconnection with family and friends and the loss of family members being risk factors for mental health among migrant populations.

Parenting was one of the family-related post-migration stressors acknowledged by some of the women, in particular, as impacting their mental health. Sole parents, in particular, were challenged to effectively parent their children, because they had no other person to support the supervision of the children at home while they worked. Pre-pandemic, they sourced some support from other African neighbors and friends. These women seemed to be unable to access childcare services, due to the difficult financial situation they faced. It was apparent that parenting had put these women into a difficult situation that forced them to reduce their work hours or to quit other activities in lieu of supervising or taking care of the children. These situations were more dire during the pandemic, as restrictions and public health measures put in place to curb the pandemic prohibited movements to other people’s homes, including friends. This was reported to lead to, in one case, a woman leaving her children in an unsafe situation with their father as she had no alternative child care arrangements when she went to work. Consistent with the findings of the current study, parenting challenges arising from different factors (insufficient job opportunities, racism, intercultural differences between parents and children) among both the skilled and refugee African migrants in Australia have been noted in previous studies [56,57,58]. Because of these parenting challenges, women reduced their working opportunities, with a subsequent loss of income which ultimately worsened the already precarious financial condition, leading to more stress and frustration. In addition to the home-related situations, concerns about poor children’s attitudes and behaviors at school led to expulsion from school, increasing the stress among some of the participants. These findings are not unique to African migrants, but support previous studies involving non-migrant parents, and have reported an association between children’s behavioral problems at schools and stress and depression among parents [59,60,61].

The variations in cultural practices between the African and Australian population were also noted as factors that led to poor mental outcomes for the children as well as the adults in these populations. It is well acknowledged that in Africa, the children are raised by the community and that the elders within communities have a significant role in raising the community’s children [57]. The communal parenting practice in Africa (as opposed to the private-dominated parenting practice) also seemed to influence someof the participants’ or parents’ views of parenting. Consistent with previous studies by Mwanri and colleagues [57] and Gatwiri and colleagues [56], it is apparent that such practices are not applied in the Australian context, leading to some of the parents experiencing difficulties in the effective supervision of children at home. Similarly, the participants’ African cultural norms, that do not allow unmarried young boys and girls to live together, were indicated to cause parent–child conflicts and cause stress to some of the parents whose children chose to do so. Such conflicts and stresses that the parents face may also be an indication of the differentials in acculturation between parents and children, as they settle in their new host nation [62].

Consistent with the findings of previous studies [9,11,12,39], the current study suggests that economic or financial hardships were among the post-migration stressors for mental health challenges faced by both the female and male participants. Such economic or financial difficulties seemed to be mainly influenced by their individual inabilities to secure meaningful employment, due to various reasons such as lack of work experience in Australia, the language barrier and poor physical health. It could also be argued that, due to the language barrier, African migrants may have less access to information regarding job opportunities, which prevents their chances in job applications. As has been reported elsewhere [26], African immigrants in Australia face challenges to their labor market success, which may have an influence on their ability to fulfil their necessities and family needs, causing psychological burden or stress and depression. Similar findings have also been noted in other studies [12,39], reporting employment difficulties as risk factors for mental illness among the migrant populations. The current findings provide new evidence on the economic-related factors, suggesting that having loans and mortgages impacted on the participants’ mental health. Coupled with unemployment or job instability, loans and mortgages increased their economic burden, due to the pressure of the requirements in servicing the bank loans and mortgages.

Our findings indicate that some of female participants experienced heavier burdens and mental health challenges resulting from a combination of stressors, such as separation from spouses, difficulties in dividing time between children and work and financial hardship. This adds further evidence to the previous findings reported elsewhere [11,12,16,18,63], which have suggested such post-migration stressors are particularly faced by the female migrant populations in different parts of Australia, and other settings globally. Difficulties in socializing with other people, self-isolation and loneliness were also post-migration stressors that negatively impacted on the mental health experienced by the female African migrants. This extends the findings of a recent study with female migrants resettling in Tasmania, Australia [63], reporting loneliness and isolation as contributing factors to mental distress among these women. This study also shows the negative impact of COVID-19 on the participants’ spousal relationships, which supports the findings of previous studies with migrant and general populations, suggesting that the COVID-19 pandemic has caused more difficulties, strains and dissatisfaction with their spousal and familial relationships, which are considered as sources of the mental health challenges facing them [64,65]. Our findings also report on the adverse influence of the COVID-19 pandemic in migrants’ employment, which is in line with the previous studies reporting difficulties faced by the migrant populations in many countries to find or secure a job during pandemic [66,67,68].

Our findings indicate that communal characteristics and strong ties with the African communities are strengths that can be utilized in the response to the mental health challenges faced by African community members. It is also worth noting that the collective communities, community structures and interactions provide the members of the communities with common traits that build resilience through ideas, experiences, skills and knowledge [69,70]. These characteristics have been reported as assisting individuals, families and communities to overcome shocks and stresses [71]. These are rooted in their tradition, within which they share common values, norms, beliefs, aspirations and cooperatively organize many aspects of their lives in communities [72,73]. Thus, the empowerment and encouragement of African community members to engage in mental health service delivery are important, as organizations can then provide socially and culturally acceptable services. The current findings add further evidence to the various reported strategies used by migrant populations to cope with the mental health challenges facing them in Australia and other host countries. These include the use of religious beliefs and reliance on God, cognitive strategies, such as reframing the situation, relying on their inner resources and focusing on future wishes and aspirations, and social support-seeking, problem-solving and reliance on family [74,75,76,77].

Finally, this study has some strengths, as well as limitations similar to those that have been reported elsewhere [44,45], and which should be considered in interpreting the current findings. The accounts reported in this paper are important because they provide important insights into the post-migration stressors that are precursors to the important mental health issues among these populations, which affect successful resettlement and integration. However, it is important to note that the involvement of a small number of African migrants in South Australia may have led to a biased overview of the views and experiences of African migrants about the risk factors for the mental health issues they face. While the stories of the participants in the current study are important, they do not represent the general experiences of all of the Africans in Australia. The use of digital platforms, such as WhatsApp/Zoom, due to COVID-19 restrictions may have also led to the possibility of excluding those African migrants who did not have access to these digital platforms. However, the use of qualitative design and qualitative data analysis framework was a strength, as previously reported [44,45].

## 5. Conclusions

This study highlights several post-migration stressors that impact on mental health among African migrants in South Australia. The findings suggest that family-related issues, such as separation from family members, spousal separation and difficulties in parenting, including effectively managing their child’s behaviors at school, were mental health stressors. Accommodating the African cultural practices of elders and communal parenting, including disciplining the children, which were not supported in Australia, was stressful for some of the participants. The cultural norms, that do not allow young unmarried people to live together, were also factors that caused child–parent conflict and stress on parents. In addition, poor economic conditions related to the limited employment opportunities led to poor fulfilment of family needs, including servicing loans and mortgages, which further impacted on the mental status of the participants. The findings indicate the need for social support, such as subsidized or free childcare services for African communities which could help them, especially to manage their schedule for kids and work. In addition, there is a need for assistance for African communities to navigate the variations between the socio-cultural norms in Australia and their own cultural practices. Finally, as economic or financial hardships were a main risk factor for the mental health challenges facing them, future studies exploring what needs to be achieved by the government and non-governmental institutions to support the access of the African migrant population to jobs are recommended.

## Data Availability

The data presented in this study are available on request from the corresponding author. The data are not made publicly available due to restrictions set by the human research ethics committee.

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
