# Peer review of "Post-Migration Stressors and Mental Health for African Migrants in South Australia: A Qualitative Study"

_ijerph, 2022, doi:10.3390/ijerph19137914_

Round 1
Reviewer 1 Report
The abstract is too long, the conclusions should be more concrete and the wording should be revised.
The methodology is not well described, which does not allow us to analyse the scientific validity of the results obtained, and the ethical principles of confidentiality are not contemplated, as it is not clear that informed consent was requested from the participants or that the design of the study was endorsed by an ethics committee.
The article addresses the stressors affecting the migrant population after settlement, but although mental health is mentioned, there is no mention of the types of disorders they have suffered and have been in demand, so this part is not well addressed in the work.
The introduction does not state the justification for the need to carry out the study, it merely provides data from similar studies carried out with similar results to those obtained, and therefore does not make any new scientific contribution.
Author Response
Reviewer 1:
The abstract is too long, the conclusions should be more concrete and the wording should be revised.
Response:
- Abstract has been revised
We conducted a qualitative study involving African migrants (n=20) and service providers (n=10) in South Australia to explore mental health stressors, access to mental health services and how to improve mental health services for African migrant populations. This paper presents the views and experiences of African migrants about post-migration stressors they faced in resettlement that pose mental health challenges. Participants were recruited using the snowball sampling technique. To align with the COVID-19 pandemic protocol, data collection was conducted using one-on-one online interviews through Zoom or WhatsApp video calls. Data analysis was guided by the framework analysis. Post-migration stressors including separation from family members and significant others, especially spouses, imposed significant difficulties to care provision and in managing children’s attitudes and behaviours-related troubles at school. African cultural practices involving the community especially elders in care provision and disciplining children, were not consistent with Australian norms, compounding mental health stressors for all involved. The African cultural norms that do not allow young unmarried people to live together, also contributed to child-parent conflicts, enhancing parental mental stressors. Additionally, poor economic conditions and employment-related difficulties were post migration stressors that participants faced. The findings indicate the need for policy and intervention programs that address the above challenges. The provision of interventions including social support such as subsidised or free childcare services could help leverage their time and scheduled paid employment, creating time for effective parenting and to improve their mental health and wellbeing. Future studies exploring what needs to be done by government and non-governmental institutions to support enhanced access to social and employment opportunities for the African migrant population are also recommended.
The methodology is not well described, which does not allow us to analyse the scientific validity of the results obtained, and the ethical principles of confidentiality are not contemplated, as it is not clear that informed consent was requested from the participants or that the design of the study was endorsed by an ethics committee.
Response:
- We provide a brief overview of the methods as details have been previously reported elsewhere [44, 45].
We have also added more explanation to the methodological section to make it clear.
2.1. Study design and participant recruitment
We used a qualitative study design to explore the views and experiences of 20 African migrants and 10 service providers in South Australia regarding issues related to post-migration stressors for mental health challenges, barriers to accessing mental health services and strategies to improve access of African migrants to the services [44, 45]. The use of qualitative design in this study assisted us to explore in-depth participants’ views, experiences, understanding and interpretations about various factors that influenced their mental health condition after their arrival in Australia. Out of this broader study, the current paper presents the perspectives and experiences of the 20 African migrants regarding post-migration stressors affecting mental health.
The recruitment used a snowball sampling technique. The researchers started the recruitment by distributing the study information packs to potential participants through community groups or associations, and organisations providing services for migrant populations in South Australia. The information packs contained contact details of the field researcher. Potential participants who received the information packs personally contacted the field researcher to confirm their participation and were recruited for an interview. Following the interviews of the initial participants, the snowball sampling technique was also employed by asking them to disseminate the information packs to their families or eligible friends and colleagues who might be willing to take part in this study. This procedure was used throughout the recruitment process, and finally, 20 African migrants were recruited and interviewed. No participants withdrew from the study prior to or during the interviews. None of the participants were known to the researchers prior to the recruitment and interviews. The inclusion criteria used to recruit the participants were (i) one had to be an African migrant, (ii) aged 18 years old and above (community participants), or (iii) a staff at an institution or organisation providing services for migrants and/or refugees (for service providers).
Of the 20 African migrants included in the study (half female and male), the age interval ranged from 18 to 60 years old, with the majority being aged 41 to 60 years (n=12 people). They were originally from eight different African countries namely: The Democratic Republic of Congo, South Sudan, Liberia, Sierra Leone, Burundi, Ethiopia, Kenya and Somalia. They were self-identified as African migrants. The duration of their living in Australia ranged from 3 to 20 years. The majority of the participants (n=11) engaged in different types of paid jobs, while the other (n=7) were currently unemployed or actively searching for a job, and the rest (n=2) were full- time university students. The participants are of mixed background, with the majority being immigrants, a couple being refugees and none being an asylum seeker. They are referred to as migrants as our focus was mainly on post-resettlement stressors for mental health challenges facing them regardless of their status or how they arrived in Australia.
2.2 Data collection and analysis
One-on-one in-depth interviews were carried out from May to October 2020 to collect data from the participants. Interviews were conducted using online platforms (Zoom or WhatsApp video calls), which were mutually agreed upon by participants and field researcher due to Covid-19 related protocols and restrictions. The duration of the interviews ranged from 40 to 60 minutes. Interviews were recorded using a digital recorder and conducted in English. Prior to the interviews, each participant was asked whether they required an interpreter to assist them but none of them request for an interpreter. The participants could speak English and well engaged the conversation during the interviews. Fieldnotes were also sometimes undertaken by the interviewer, if felt necessary, and then integrated into each transcript during the transcription process. Interviews ceased once no new added information or response was provided by the last few participants, which was an indication of data saturation. All of the interviews were conducted by a female researcher from a non-African background who has significant experience in conducting research activity in these populations. There was not repeated interview with any participants. At the end of each interview, the interviewer offered an opportunity for each participant to read and comment the interview transcript after transcription, but none requested to do so.
Data were transcribed verbatim, and analysis was guided by five steps of qualitative data analysis framework introduced in Ritchie and Spencer [46, 47], which include: (i) familiarisation with the data. This was carried out through reading repeatedly reading the transcript of each interview, breaking down information into small chunks of information or data extracts, and highlighting and giving comments and labels to those data extracts; (ii) identification of a thematic framework through which the recurrent and emerging key ideas and concepts regarding the topic under researched from each interview transcript were identified and listed. These ideas, issues and concepts were then used to develop the thematic framework. The thematic framework was identified and developed deductively based on prior knowledge, and inductively based on the emerging themes; (iii) Indexing the data or coding, which was started by creating open codes to data extracts that had been created in the previous steps, which resulted in a collection of a long list of codes. Following this, close coding was carried out to identify and collate similar codes to reduce the number codes to a manageable number. Also, different codes that formed a pattern were grouped under the same theme or sub-theme; (iv) creating a chart for the data through reorganizing the themes and their data extracts that had been created and coded in previous steps in a summary of chart which enabled data comparison within each interview and across interviews; and (v) mapping and interpretation of the data entirely [44, 45]. These steps were followed to support the management of these qualitative data in a structured way and enhance the rigour and transparency of the analytical process [46, 48, 49]. We have reported other findings from the study elsewhere [44, 45].
- Each participant signed an informed consent form to confirm their willingness to participate, and they returned the consent form to the interviewer prior to the commencement of the interviews. This study was approved by Social Science and Behaviour Research Ethics Committee, Flinders University (No. 8570). The details about ethics approval and consent from participants have been previously reported elsewhere [44, 45].
The introduction does not state the justification for the need to carry out the study, it merely provides data from similar studies carried out with similar results to those obtained, and therefore does not make any new scientific contribution.
Response:
- We have revised and made the knowledge gap clear:
Despite a range of risk factors for mental health issues among migrant populations and refugees globally and in the context of Australia as reported in the aforementioned studies, there is a lack of studies and evidence on how post-settlement factors such as spousal relationships, children-related issues, culture practices of raising children, and issues related to mortgage, loans and bills, influence mental health outcomes of migrant populations in host communities. This paper aimed to fill in this gap in knowledge and focused on exploring post-settlement stressors for mental health issues among African migrants in South Australia. In particular the attention is on family and community relationships, parenting and financial matters. Understanding these factors can be useful for the development of policy and intervention programs to support migrant populations’ resettlement and integration in Australia and other settings globally.
The article addresses the stressors affecting the migrant population after settlement, but although mental health is mentioned, there is no mention of the types of disorders they have suffered and have been in demand, so this part is not well addressed in the work.
Response:
- Types of mental health challenges have been provided in our synthesis and highlighted in the quotes or narratives of the participants throughout results section.
Reviewer 2 Report
The manuscript addresses the challenges and stressors, as well as the resources of African migrants in South Australia during the post-migration period. The findings indicate family-related issues, cultural norms, and poor economic conditions as the main stressors that these immigrants cope with during this period. In addition, the study highlights the community as a significant resource that assists immigrants in coping with these stressors. The research findings contribute to pinpointing the needs of immigrants and can also assist in developing services to address those needs, such as subsidized or free childcare services, as the researchers suggest. However, despite the importance of the study, it is not clear what the unique contribution of the research and its innovation is, beyond a large body of research knowledge that has already examined these issues. In addition, the Methods chapter and the Findings chapter require further tightening, as detailed below:
1. The introduction is clear, well-written, and well-structured, as it provides a comprehensive overview of the characteristics of immigration and the challenges and difficulties which immigrants face. However, in light of the extensive literature review, the rationale behind the research is not clear enough, nor are its uniqueness and its contribution to the broad body of existing research knowledge on the subject. (Lines 99-105).
2. In the sub-chapter "Data collection and analysis", it is written that the interviews were conducted in English (Line 133). However, it might be assumed that for some of the interviewees, English was not their mother tongue, and as a result, they might have had difficulty expressing themselves, or even understanding the interviewer. It is recommended to provide more information regarding this language barrier and the strategies the interviewer and researchers used to overcome it.
3. "The five steps of qualitative data analysis framework" (Lines 137-140), seems very interesting and relevant for analyzing the qualitative data of the research. A more detailed description of this analysis method as well as a clearer connection to the process of analyzing the findings are recommended.
4. The last sentences of the sub-chapter "Data collection and analysis" (Lines 143-147) are not clear. Do these two issues emerge from analyzing the data? How are they related to the research findings? There is an inconsistency between these two themes and the research findings described in the following chapter.
5. The description of the themes in the Results chapter appears at the end of the sub-chapter "Participants’ sociodemographic information" (Lines 157-159). This description should appear at the beginning of the chapter or after the sociodemographic information sub-chapter, and not in it. In addition, the fourth theme "Community strengths" is missing in this general description of the themes.
6. In the Results chapter, the effect of the COVID-19 pandemic on the stressors that the immigrants coped with is mentioned in the quotes of the interviewees in the theme "Spousal relationships and children related issues" (Lines 211-220), in the theme "Economic factors" (Lines 309-313), and in the theme "Community strengths" (Lines 365-369). However, the reference to the pandemic in the research findings is limited. It is recommended to address the effect of the pandemic on the immigrants' stressors more directly in the Results chapter as well as in the Discussion chapter.
7. In the Discussion chapter, it is recommended to better clarify why breakups are taboo for African immigrants (Lines 385-392).
Author Response
Reviewer 2
The manuscript addresses the challenges and stressors, as well as the resources of African migrants in South Australia during the post-migration period. The findings indicate family-related issues, cultural norms, and poor economic conditions as the main stressors that these immigrants cope with during this period. In addition, the study highlights the community as a significant resource that assists immigrants in coping with these stressors. The research findings contribute to pinpointing the needs of immigrants and can also assist in developing services to address those needs, such as subsidized or free childcare services, as the researchers suggest. However, despite the importance of the study, it is not clear what the unique contribution of the research and its innovation is, beyond a large body of research knowledge that has already examined these issues. In addition, the Methods chapter and the Findings chapter require further tightening, as detailed below:
The introduction is clear, well-written, and well-structured, as it provides a comprehensive overview of the characteristics of immigration and the challenges and difficulties which immigrants face. However, in light of the extensive literature review, the rationale behind the research is not clear enough, nor are its uniqueness and its contribution to the broad body of existing research knowledge on the subject. (Lines 99-105).
Response:
- Despite a range of risk factors for mental health issues among migrant populations and refugees globally and in the context of Australia as reported in the aforementioned studies, there is a lack of studies and evidence on how post-settlement factors such as spousal relationships, children-related issues, culture practices of raising children, and issues related to mortgage, loans and bills, influence mental health outcomes of migrant populations in host communities. This paper aimed to fill in this gap in knowledge and focused on exploring post-settlement stressors for mental health issues among African migrants in South Australia. In particular the attention is on family and community relationships, parenting and financial matters. Understanding these factors can be useful for the development of policy and intervention programs to support migrant populations’ resettlement and integration in Australia and other settings globally.
In the sub-chapter "Data collection and analysis", it is written that the interviews were conducted in English (Line 133). However, it might be assumed that for some of the interviewees, English was not their mother tongue, and as a result, they might have had difficulty expressing themselves, or even understanding the interviewer. It is recommended to provide more information regarding this language barrier and the strategies the interviewer and researchers used to overcome it.
Response:
- One-on-one in-depth interviews were carried out from May to October 2020 to collect data from the participants. Interviews were conducted using online platforms (Zoom or WhatsApp video calls), which were mutually agreed upon by participants and field researcher due to Covid-19 related protocols and restrictions. The duration of the interviews ranged from 40 to 60 minutes. Interviews were recorded using a digital recorder and conducted in English. Prior to the interviews, each participant was asked whether they required an interpreter to assist them but none of them request for an interpreter. The participants could speak English and well engaged the conversation during the interviews. Fieldnotes were also sometimes undertaken by the interviewer, if felt necessary, and then integrated into each transcript during the transcription process. Interviews ceased once no new added information or response was provided by the last few participants, which was an indication of data saturation. All of the interviews were conducted by a female researcher from a non-African background who has significant experience in conducting research activity in these populations. There was not repeated interview with any participants. At the end of each interview, the interviewer offered an opportunity for each participant to read and comment the interview transcript after transcription, but none requested to do so.
"The five steps of qualitative data analysis framework" (Lines 137-140), seems very interesting and relevant for analyzing the qualitative data of the research. A more detailed description of this analysis method as well as a clearer connection to the process of analyzing the findings are recommended.
Response:
- Data were transcribed verbatim, and analysis was guided by five steps of qualitative data analysis framework introduced in Ritchie and Spencer [46, 47], which include: (i) familiarisation with the data. This was carried out through reading repeatedly reading the transcript of each interview, breaking down information into small chunks of information or data extracts, and highlighting and giving comments and labels to those data extracts; (ii) identification of a thematic framework through which the recurrent and emerging key ideas and concepts regarding the topic under researched from each interview transcript were identified and listed. These ideas, issues and concepts were then used to develop the thematic framework. The thematic framework was identified and developed deductively based on prior knowledge, and inductively based on the emerging themes; (iii) Indexing the data or coding, which was started by creating open codes to data extracts that had been created in the previous steps, which resulted in a collection of a long list of codes. Following this, close coding was carried out to identify and collate similar codes to reduce the number codes to a manageable number. Also, different codes that formed a pattern were grouped under the same theme or sub-theme; (iv) creating a chart for the data through reorganizing the themes and their data extracts that had been created and coded in previous steps in a summary of chart which enabled data comparison within each interview and across interviews; and (v) mapping and interpretation of the data entirely [44, 45]. These steps were followed to support the management of these qualitative data in a structured way and enhance the rigour and transparency of the analytical process [46, 48, 49]. We have reported other findings from the study elsewhere [44, 45].
The last sentences of the sub-chapter "Data collection and analysis" (Lines 143-147) are not clear. Do these two issues emerge from analyzing the data? How are they related to the research findings? There is an inconsistency between these two themes and the research findings described in the following chapter.
Response:
- This sentence has been deleted
The description of the themes in the Results chapter appears at the end of the sub-chapter "Participants’ sociodemographic information" (Lines 157-159). This description should appear at the beginning of the chapter or after the sociodemographic information sub-chapter, and not in it. In addition, the fourth theme "Community strengths" is missing in this general description of the themes.
Response:
- Thank you very much for the detailed observation, this part has been revised as suggested and sociodemographic information of the participants has been integrated in “Study design and participant recruitment” section.
In the Results chapter, the effect of the COVID-19 pandemic on the stressors that the immigrants coped with is mentioned in the quotes of the interviewees in the theme "Spousal relationships and children related issues" (Lines 211-220), in the theme "Economic factors" (Lines 309-313), and in the theme "Community strengths" (Lines 365-369). However, the reference to the pandemic in the research findings is limited. It is recommended to address the effect of the pandemic on the immigrants' stressors more directly in the Results chapter as well as in the Discussion chapter.
Response:
- The effect of COVID-19 pandemic on these aspects have been made clear in these result sections and discussion of these impacts have been provided in discussion section:
Our findings indicate that some female participants experienced heavier burdens and mental health challenges resulting from a combination of stressors such as separation from spouses, difficulties to divide time for children and work, and financial hardship. This adds further evidence to the previous findings reported elsewhere [11, 12, 16, 18, 63] which have suggested such post-migration stressors are particularly faced by female migrant populations in different parts of Australia and other settings globally. Difficulties in socialising with other people, self-isolation and loneliness were also post-migration stressors that negatively impacted mental health experienced by female African migrants. This extends the findings of a recent study with female migrants resettling in Tasmania, Australia [63], reporting loneliness and isolation as contributing factors to mental distress among these women. This study also shows the negative impact of COVID-19 on participants’ spousal relationships, which supports the findings of previous studies with migrant and general populations suggesting that COVID-19 pandemic has caused more difficulties, strains and dissatisfaction on their spousal and familial relationships, which are considered as sources of mental health challenges facing them [64, 65]. Our findings also report of adverse influence of COVID-19 pandemic in migrant’s employment, which is in line with the previous studies reporting difficulties faced by migrant populations in many countries to find or secure a job during pandemic [66–68].
In the Discussion chapter, it is recommended to better clarify why breakups are taboo for African immigrants (Lines 385-392).
Response:
- Thank you very much for the detailed observation: this has been revised.
Reviewer 3 Report
Strength of the theoretical framework that collects the most important problems during the migration process (settlement stage at destination linked to triggering factors of psychosocial risks sources of different models of stress and subsequent disorders and / or diseases related to mental health. The bibliography in this regard is adequate and up to date.
I would like to highlight the focus of the study and methodology, which makes it possible to include the perception of the group under study, which is necessary and scarce in this field.
The methodological design is correct and the sample is adequate, as it is a qualitative study using semi-structured interviews. It would have been desirable to contrast the responses with the service operators or managers, analysing their level of agreement/agreement with the statements made by the migrants.
It also highlights the value of the results, expected in view of previous research, but absolutely necessary because they do not respond to a theoretical approach but to experience, to the experience of the group. Following the line set by the Sustainable Development Goals (Agenda 20-30) and the Global Compact on Migration (UN 2018), it gives meaning to the possibilities of intervention based on the participation of the group affected, from their own particular experiences.
It is a current and necessary work
Author Response
Reviewer 3
Strength of the theoretical framework that collects the most important problems during the migration process (settlement stage at destination linked to triggering factors of psychosocial risks sources of different models of stress and subsequent disorders and / or diseases related to mental health. The bibliography in this regard is adequate and up to date.
Response:
- Thank you very much.
I would like to highlight the focus of the study and methodology, which makes it possible to include the perception of the group under study, which is necessary and scarce in this field.
The methodological design is correct and the sample is adequate, as it is a qualitative study using semi-structured interviews.
Response:
- Thank you very much.
It would have been desirable to contrast the responses with the service operators or managers, analysing their level of agreement/agreement with the statements made by the migrants.
Response:
- Thank you very much for this excellent observation. Unfortunately, we didn’t explore the views of service providers on post-migration stressors for mental health challenges facing immigrants. Interviews with the service providers were mainly focused on their views and experiences regarding barriers to access to mental health services and strategies to improve access to the services by migrant populations.
It also highlights the value of the results, expected in view of previous research, but absolutely necessary because they do not respond to a theoretical approach but to experience, to the experience of the group. Following the line set by the Sustainable Development Goals (Agenda 20-30) and the Global Compact on Migration (UN 2018), it gives meaning to the possibilities of intervention based on the participation of the group affected, from their own particular experiences.
It is a current and necessary work
Response:
- Thank you very much for your appreciation of this work.
Reviewer 4 Report
This article sheds further light into the set of issues that make the aftermath of resettlement problematic for many immigrants and refugees in destination countries. The fact that the paper is focused on immigrants and refugees from Africa in Australia provides a valuable context and the authors do a good job showing how worries about family left behind, lack of communal support, domestic violence and spousal issues and unemployment and financial difficulties can lead to negative mental health outcomes in vulnerable displaced populations. My concerns with the article are as follows:
1) There is no explanation as to why people from 8 different countries are categorized together as "African" immigrants: Is that because there is a racial discrimination? Or because these immigrants identify themselves as belonging to an "African" group? Or are we simply using that category without problematizing it? I think a bit of an explanation might be necessary.
2) The background of the surveyed population is not well explained. There is a huge age gap between respondents, and most seem to be women. It is unclear which ones were refugees and which ones were immigrants. Those two categories entail very different journeys. It is unclear if the respondents were asylum seekers and through which process they came into the country. Given Australia's problematic policies in sea vs land arrivals, it would have helped to clarify some of these points. It is also important to show the differences between those who were resettled refugees vs. immigrants. There is also no socio-economic data in terms of jobs, sectors, education levels and skills. Qualitative data is stronger when such details are provided.
3) Given that many of the respondents are women, there is a gendered perspective missing in the article. I think issues such as domestic violence is important but the gendered context needs to be specified in the literature review and in the overall analysis.
4) When we work with refugees and immigrants, it is important to recognize their agency and resilience and ability to cope. This article starts out by painting these populations somewhat more prone to mental health problems, comparing them to Australians. If one were to study Australians who have been through rough times such as displacement, homelessness, poverty and violence, one could see that mental health issues are not unique to immigrant populations. There is little discussion of how the respondents cope with their circumstances and feelings, except a few references to community events and festivals. There needs to be more scholarly research about the resilience and coping mechanisms of these populations rather than pathologizing them simply because they are refugees or immigrants. I don't think this is the intent of the authors, but the lack of references to immigrant and refugee agency and resilience are noteworthy.
If the authors are able to address these issues, I would recommend this article for publication.
Author Response
Reviewer 4
This article sheds further light into the set of issues that make the aftermath of resettlement problematic for many immigrants and refugees in destination countries. The fact that the paper is focused on immigrants and refugees from Africa in Australia provides a valuable context and the authors do a good job showing how worries about family left behind, lack of communal support, domestic violence and spousal issues and unemployment and financial difficulties can lead to negative mental health outcomes in vulnerable displaced populations. My concerns with the article are as follows:
There is no explanation as to why people from 8 different countries are categorized together as "African" immigrants: Is that because there is a racial discrimination? Or because these immigrants identify themselves as belonging to an "African" group using that category without problematizing it? I think a bit of an explanation? Or are we simply might be necessary.
Response:
- The information packs contained contact details of the field researcher. Potential participants who received the information packs personally contacted the field researcher to confirm their participation and were recruited for an interview.
- They were self-identified as African migrants.
The background of the surveyed population is not well explained. There is a huge age gap between respondents, and most seem to be women. It is unclear which ones were refugees and which ones were immigrants. Those two categories entail very different journeys. It is unclear if the respondents were asylum seekers and through which process they came into the country. Given Australia's problematic policies in sea vs land arrivals, it would have helped to clarify some of these points. It is also important to show the differences between those who were resettled refugees vs. immigrants. There is also no socio-economic data in terms of jobs, sectors, education levels and skills. Qualitative data is stronger when such details are provided.
Response:
- As the study was based on voluntary and willingness to participation, it does not represent the targeted population, rather provides rich and detailed data of experiences of the participants. The recruitment was also through snowball methods which makes it difficult to estimate how many people received the information but did not volunteer to participate. Of the 20 African migrants included in the study (half female and male), the age interval ranged from 18 to 60 years old, with the majority being aged 41 to 60 years (n=12 people). They were originally from eight different African countries namely: The Democratic Republic of Congo, South Sudan, Liberia, Sierra Leone, Burundi, Ethiopia, Kenya and Somalia. They were self-identified as African migrants. The duration of their living in Australia ranged from 3 to 20 years. The majority of the participants (n=11) engaged in different types of paid jobs, while the other (n=7) were currently unemployed or actively searching for a job, and the rest (n=2) were full- time university students. The participants are of mixed background, with the majority being immigrants, a couple being refugees and none being an asylum seeker. They are referred to as migrants as our focus was mainly on post-resettlement stressors for mental health challenges facing them regardless of their status or how they arrived in Australia.
Given that many of the respondents are women, there is a gendered perspective missing in the article. I think issues such as domestic violence is important but the gendered context needs to be specified in the literature review and in the overall analysis.
Response:
- As we have presented in the methods section, the respondents were half male and half female.
- We provided some findings specific to women, which are not identified in interviews with male participants:
Spousal relationship issues were indicated to negatively impact some female participants’ mental health outcomes. For example, three female participants reported being abandoned by their husband or fiancé. Their expectations for their own families and relationships seemed to be ruined by such separation or break up which further caused them various negative psychological challenges such as stress and feeling of loneliness. It also seemed that COVID-19 situation and the enforcement of the preventive protocols may have corroborated these challenging mental health issues as they faced. The following narrative of a woman who has been living in South Australia for more than ten years illustrates these assertions:
“I have a lot of problems with my health. Especially, right now I’m sitting – I’m talking with you as a human, but I have a deep thing in my heart that I really need to let out. Since the COVID-19, it has been disturbing me a lot. And can you imagine, you came with a family and because of money or some kind of …. your husband just left you all alone by yourself with the kids, yeah. And it has been bothering me a lot. .... And right now personally as I’m talking with you, I have a counsellor who’s supposed to be meeting me tomorrow, so we can talk about it. Because I’m stressed out. Can you imagine you expect your family to be a certain way, and things are not working out for you? Like, for me, I got married back home, and all my family knows that I’m with my husband and I came here and he’s no more with me. It stresses me out a lot” (P16, 54 years old).
Spousal relationship issues seemed to have consequences on participants’ responsibility to raise their children. Similar stories were echoed by other women who were sole parents who found it difficult to raise children on their own. For example, a female participant described how her children had experienced violence from their father which led to their separation, but because of the lack of resources or support, she at times had to seek his support to look after their children as she needed to work. This situation was described as painful to her as she did not want to put her children into such a risky environment:
“We (the woman and her husband) used to live together, but unfortunately we separated due to violence. So right now they (my children) live with me but sometimes I take them to their dad when I’m going to work because nobody’s there to look after them and I don’t want to make other people feel like I am bothering them with my kids. The fact that we separated, I normally just say - it pains me a lot to take them (my children) back to the environment, because he (my former husband) drinks a lot and it’s not safe for them, it pains me a lot. But at the same time, I don’t have any other options. I moved out from everything, but then the kids make me go back” (P11, 37 years old).
We have also added some literature on this in our analysis and discussion of this aspect:
Our findings indicate that some female participants seemed to experience heavier burdens and mental health challenges resulting from a combination of stressors such as separation from spouses, difficulties to divide time for children and work, and financial hardship. This adds further evidence to the previous findings reported elsewhere [11, 12, 16, 18, 63] which have suggested such post-migration stressors are particularly faced by female migrant populations in different parts of Australia and other settings globally. Difficulties in socialising with other people, self-isolation and loneliness were also post-migration stressors that negatively impacted mental health experienced by female African migrants. This extends the findings of a recent study with female migrants resettling in Tasmania, Australia [63], reporting loneliness and isolation as contributing factors to mental distress among these women. This study also shows the negative impact of COVID-19 on participants’ spousal relationships, which supports the findings of previous studies with migrant and general populations suggesting that COVID-19 pandemic has caused more difficulties, strains and dissatisfaction on their spousal and familial relationships, which are considered as sources of mental health challenges facing them [64, 65]. Our findings also report of adverse influence of COVID-19 pandemic in migrant’s employment, which is in line with the previous studies reporting difficulties faced by migrant populations in many countries to find or secure a job during pandemic [66–68].
When we work with refugees and immigrants, it is important to recognize their agency and resilience and ability to cope. This article starts out by painting these populations somewhat more prone to mental health problems, comparing them to Australians. If one were to study Australians who have been through rough times such as displacement, homelessness, poverty and violence, one could see that mental health issues are not unique to immigrant populations. There is little discussion of how the respondents cope with their circumstances and feelings, except a few references to community events and festivals. There needs to be more scholarly research about the resilience and coping mechanisms of these populations rather than pathologizing them simply because they are refugees or immigrants. I don't think this is the intent of the authors, but the lack of references to immigrant and refugee agency and resilience are noteworthy.
Response:
- Thanks for this detailed observation and comments, we have deleted that statement.
- We have consulted various coping strategies used by migrant populations as reported in previous studies and provided them in the discussion section:
Our findings indicate that communal characteristics and strong ties with African communities are strengths that can be utilised in the response to mental health challenges faced by African community members. It is also worth noting that as collective communities, community structures and interactions provide members of communities common traits that build resilience through ideas, experiences, skills and knowledge [69, 70]. These characteristics have been reported to assist individuals, families and communities to overcome shocks and stresses [71]. These are rooted in their tradition within which they share common values, norms, beliefs, aspirations and cooperatively organise many aspects of their lives in communities [72, 73]. Thus, empowerment and encouragement of African community members to engage in mental health service delivery are important as they can provide socially and culturally acceptable services. The current findings add further evidence to various reported strategies used by migrant populations to cope mental health challenges facing them in Australia and other host countries. These include the use of religious beliefs and reliance on God, cognitive strategies such as reframing the situation, relying on their inner resources and focusing on future wishes and aspirations, and social support-seeking, problem-solving, and reliance on family [74-77].
Round 2
Reviewer 1 Report
This text is repeated twice: All interviews were conducted by a researcher of non-African origin who has significant experience in conducting research activities with these populations. No participant was re-interviewed. At the end of each interview, the interviewer provided an opportunity for each participant to read and comment on the interview transcript after transcription, but none requested to do so. All interviews were conducted by a researcher of non-African origin who has significant experience in conducting research activities with these populations. No participant was re-interviewed. At the end of each interview, the interviewer provided an opportunity for each participant to read and comment on the interview transcript after transcription, but none requested to do so.
The authors have amended the above corrections very well so that in my opinion they can be accepted for publication.
Reviewer 2 Report
All suggestions were adequately addressed